# A Barter Economy in Tumors: Exchanging Metabolites through Gap Junctions

**DOI:** 10.3390/cancers11010117

**Published:** 2019-01-20

**Authors:** Pawel Swietach, Stefania Monterisi

**Affiliations:** Department of Physiology, Anatomy & Genetics, University of Oxford, Oxford OX1 3PT, UK; stefania.monterisi@dpag.ox.ac.uk

**Keywords:** connexins, diffusion, cancer, PDAC, CRC, pH homeostasis, warburg effect, HCO_3_^−^

## Abstract

To produce physiological functions, many tissues require their cells to be connected by gap junctions. Such diffusive coupling is important in establishing a cytoplasmic syncytium through which cells can exchange signals, substrates and metabolites. Often the benefits of connectivity become apparent solely at the multicellular level, leading to the notion that cells work for a common good rather than exclusively in their self-interest. In some tumors, gap junctional connectivity between cancer cells is reduced or absent, but there are notable cases where it persists or re-emerges in late-stage disease. Diffusive coupling will blur certain phenotypic differences between cells, which may seem to go against the establishment of population heterogeneity, a central pillar of cancer that stems from genetic instability. Here, building on our previous measurements of gap junctional coupling between cancer cells, we use a computational model to simulate the role of connexin-assembled channels in exchanging lactate and bicarbonate ions down their diffusion gradients. Based on the results of these simulations, we propose that an overriding benefit of gap junctional connectivity may relate to lactate/bicarbonate exchange, which would support an elevated metabolic rate in hypoxic tumors. In this example of barter, hypoxic cancer cells provide normoxic neighbors with lactate for mitochondrial oxidation; in exchange, bicarbonate ions, which are more plentiful in normoxic cells, are supplied to hypoxic neighbors to neutralize the H^+^ ions co-produced glycolytically. Both cells benefit, and so does the tumor.

## 1. Trading within Biological Systems

In economics, barter is the exchange of goods or services without the involvement of money, under the premise that the two sides coincidentally want each other’s offer. As a means of trading, barter is now rare, but is historically significant for its role in enabling the division of labor. In early societies, barter would typically take place between partners who were not very familiar to one another [1], perhaps because a complete and immediate transaction was a way of diffusing concerns about the partner’s trustworthiness at a time when regulated systems of credit were not available [2]. The introduction of money and credit systems transformed trading and allowed societies to accrue considerable wealth.

A biological cell hosts a collection of specialized processes that inherently demonstrate division of labor. Many of these processes require an energetic input, supplied in the form of ATP, which serves as a universally-accepted currency of energy. An analogy can be drawn to an advanced economy, where metabolism generates ATP (‘money’) to power (‘pay’) various specialized biological processes (‘crafts’), so that these collectively support survival and proliferation (‘wealth’). Such a system functions properly because it is in the interest of all constituents that the cell thrives. At the multicellular level, specialization occurs among different tissues which are ‘paid’ by substrates (e.g., glucose) to engage in activities that benefit the whole body. A mechanism of credit (involving signaling and homeostasis) is implemented to ensure that the components of the system do not seek instantaneous returns. Tumors are, however, quite different. Cancer cells have a strong individualistic survival instinct, with little scope for altruistic gestures towards other cells [3,4]; moreover, cancer cells will often parasitically exploit neighboring cells, such as the host stroma [5,6]. Nonetheless, tumors have the potential to benefit from the division of labor because they are *not* homogenous tissues. Their heterogeneity arises from genetic differences among sub-populations of cancer cells, and also from variation in micro-environmental factors, such as oxygen supply which depends on the distance from the nearest capillary [3]. Reaping the benefits of the division of labor among selfish and suspicious partners may require a primitive exchange arrangement, such as barter. The instantaneous and reciprocal exchange of assets between cancer cells is achievable through inter-cellular conduits called gap junctions, first described in the 1960s [7]. 

## 2. Gap Junctions are Conduits for Exchanging Small- to Medium-Sized Molecules

Gap junctions are assembled from the juxtaposition of two hemichannels expressed at the surface of abutting cells [8]. These hemichannels are made of connexin proteins [9] which are coded by genes belonging to a family of at least 24 members [8,10]. The pore that traverses the complete channel offers a low-resistance pathway for exchanging solutes as big as 1 kDa. Essentially all metabolites, second messengers and inorganic ions are within this range of permeating substances, which results in a functional coupling between cells. The overall strength of this coupling depends on the number of gap junctional channels connecting a pair of cells.

Cell-to-cell coupling is essential to the function of many types of tissue (e.g., the spread of electrical excitation in the heart [11]) and, not surprisingly, most cells in the body are junctionally-coupled [12,13]. Many normal epithelia express connexins and manifest cell-to-cell coupling, such as Cx26, 32, 43 in pancreatic ducts and Cx26, 32, 43, 45 in colorectal epithelium [10,14,15]. Notable exceptions to the near-ubiquitous expression of connexins are blood cells, as these are stand-alone units, and fully-developed skeletal muscle, because their contractile units must be electrically-isolated to allow selective recruitment for the purpose of force gradation [16]. Historically, this list of exceptions also included tumors, on the basis that early observations suggested a lack of connectivity [17,18,19], subsequently supported by more recent work [20], as well as evidence for a tumor suppressing effect of connexins [21,22]. Contemporary findings have challenged this dogma [10], and it is now recognized that some cancers do express connexins, at least in some stages of progression [23]. 

There will be a myriad of roles for connexins in cancer, including some that relate more to their involvement in cell-to-cell adhesion [24,25,26,27], than the provision of a meaningful solute exchange pathway. Clinical studies have not been able to ascertain the direction of correlation between connexin expression and prognosis, as both positive and negative associations have been reported [10]. In colorectal cancer, for example, one study linked high Cx26 expression with poor prognosis and lung metastasis [28], whereas a subsequent study [29] came to the opposite conclusion. On the whole, there appears to be an isoform-specific pattern in the association between expression and prognosis, with Cx26 generally indicating worse prognosis and Cx43 linked to better prognosis [10]. A limitation of the aforementioned studies is that they have not determined the extent to which connexin expression produces meaningful coupling between cancer cells, a principal phenotypic readout of connexin function. Indeed, gap junctional communication is not routinely quantified in most studies of connexins in cancer (with a few exceptions, e.g., [21,25,30,31,32,33,34] and our work [35,36]). With the availability of antibodies and primers, recent studies of connexins in cancer cell lines have tended to quantify their levels in terms of mRNA or protein, which cannot faithfully predict functional coupling. Indeed, mislocalization of connexins to regions other than cell-cell contacts has been described for Cx26 in pancreatic [37] and colorectal cancer [28], putatively. Thus, our appreciation of the physiological role of cell-to-cell diffusive coupling in cancer, particularly in vivo, is still very limited. If, however, a specific type of malignancy manifested strong cell-to-cell diffusive coupling, it would be surprising if solute exchange had no bearing on cancer physiology.

One method of quantifying the degree of coupling between cells is based on fluorescence recovery after photobleaching (FRAP). A fluorescent dye, which can freely pass gap junctions (i.e., of molecular weight lower than 1 kDa), is loaded into a confluent monolayer; an exemplar colorectal cancer (CRC) cell line is shown in Figure 1. Fluorescence in one cell is then bleached (e.g., to 50% of starting level) using a high-power laser, and the signal time course registered in the bleached cell and its direct neighbors. The recovery time course is then fitted with a mono-exponential, and the initial slope (arrow in Figure 1D) provides a rate of fluorescence recovery, d*F*/d*t*, which is related to: (*i*) permeability, *P*, (*ii*) the difference in fluorescence between bleached and neighboring cells ΔF (i.e., the diffusion gradient), (*iii*) the surface area of the bleached cell at the junction with the neighboring cells, *s*, and (*iv*) the volume of the bleached cell, *v*:d*F/*d*t* = *P* × Δ*F* × *s*/*v*(1)

The *s*/*v* ratio can be approximated by quotient *ρ* of the bleached cell’s perimeter and its area in the xy plane (this assumes constant monolayer height). Thus, permeability, in units of µm/s, is:(2)$$P=\frac{dF/dt}{\Delta F\times \rho }$$

This permeability relates, of course, to the fluorescent marker, which will be bulkier than many endogenous metabolites, signals or ions. Smaller molecules are thus expected to permeate faster by an amount determined by their size, charge and interactions with the pore area [38,39].

At first sight, solute dissipation across a syncytium of cells may appear to be the antithesis of what individual cancer cells need to compete in the process of somatic evolution because diffusion will blur differences in metabolic, signaling and ionic phenotype between connected cells. In late-stage disease, genetic instability and hyper-proliferation result in clonal heterogeneity [3], but since somatic evolution selects phenotype *not* genotype, the substrate for selection will become less diverse in connected cancer cells [4,40]. The syncytial behaviors of these units limit cell autonomy and constrain cell-to-cell variation, a fundamental determinant of evolutionary change according to game theory [41]. Indeed, mathematical models of somatic evolution routinely consider each cancer cell as a unit [4]. Considering these potential drawbacks, there must be some overriding benefit that intracellular coupling could offer tumors. A plausible beneficiary of coupling is metabolism and pH regulation, two intertwined processes that are fundamental to cancer growth and generate large fluxes of solutes [42,43]. There have been reports for metabolic cooperation between cancer cells, in which substrates are made available to those cells that cannot synthesize them [44]. More recently, we have demonstrated bulk fluxes of lactate [36] and HCO_3_^−^ ions [35] across gap junctions in spheroids grown form pancreatic ductal adenocarcinoma (PDAC) cell lines.

## 3. Glycolytic Metabolism Establishes Radial Concentration Gradients of Lactate and Bicarbonate in Solid Tumors

Cancer cells require a steady flow of ATP to service the demands placed by proliferation, invasion and metastasis. Metabolic reprogramming to a glycolytic phenotype is common in many cancer cells (Warburg effect) [42,43,45,46], although a degree of mitochondrial respiration will also take place if oxygen is adequate [43,47]. The end-product of glycolytic metabolism is the ionized form of lactic acid [48], which must be removed from the cell, particularly if there is insufficient mitochondrial capacity to metabolize lactate further in the Krebs cycle. Typically, cancer cell lines produce lactate at a rate of several millimoles per minute per liter of cell volume (mM/min) [49,50,51,52,53], but cancer cell lines can attain rates as high as 20 mM/min [54]. The sole means by which lactate can cross biological membranes is in co-transport with H^+^ ions aboard H^+^-monocarboxylate transporter (MCTs) [55,56] or, to a much lesser degree, as lactic acid crossing the lipid bilayer; either way, membrane permeation is electroneutral. Once outside the cell, lactic acid dissociates, and its ionic products diffuse towards blood capillaries to complete the process of venting (Figure 2A).

Membrane permeation and extracellular diffusion can be considered as resistors in series. For cancer cells that are deep at the hypoxic core of solid tumors, the diffusive resistance will be substantial, and no amount of MCT expression could overcome this. Rate-limiting removal of the products of glycolytic metabolism will lead to their retention at the hypoxic core. The extent of this can be simulated mathematically (Figure 2B,C). The equations used in the model are presented in the Appendix A, together with an explanation of the variables and assumptions. For illustrative purposes, we assume the extracellular space occupies 25% of tumor volume, has an intrinsic buffering capacity of 3 mM/min, and its tortuosity reduces ionic diffusivities by a factor of two ([57,58,59,60]). Thus, extracellular lactate, HCO_3_^−^ and CO_2_ diffusion coefficients were 650, 900 and 1200 µm^2^/s, respectively. A similar tortuosity is modelled for the intracellular space; and the apparent H^+^ diffusion coefficient taken as 200 µm^2^/s [61]. MCT activity in cells, radius 7 µm and intracellular intrinsic buffering capacity of 15 mM/pH [62], was set to a high level, equivalent to a permeability to lactic acid of 10^3^ µm/s, and lactic acid production fixed at 20 mM/min [54]. CO_2_/HCO_3_^−^ buffer is included in the modelling, with freely-permeating CO_2_ (permeability constant 10^3^ µm/s) [57], and to facilitate its chemical reactions, exofacial carbonic anhydrase (CA) activity increased spontaneous CO_2_ hydration rate by a factor of 10 (e.g., CAIX/CAXII) [59,60]. The geometry of the modelled tissue is assumed spherical of radius 250 µm to encompass both normoxic regions at the periphery and hypoxic areas at the core. Concentrations at the boundary are set to levels normally found in blood, i.e., CO_2_ at 1.2 mM, HCO_3_^−^ at 24 mM, hence extracellular pH (pH_e_) at 7.4, and negligible lactate. In the model, cells at the tissue boundary were able to fully metabolize lactate (i.e., intracellular lactate is set to nil) and import HCO_3_^−^ ions by active transport (i.e., intracellular [HCO_3_^−^] set at 13.5 mM).

Lactic acid produced glycolytically by cancer cells (‘source’) is vented to the boundary of the tissue (‘sink’), down the route of least resistance. Essentially all traffic involves permeation across the membrane of the glycolytic cell, followed by a meandering across the extracellular compartment. Results of the simulation indicate substantial gradients of lactate in both the intra- and extracellular spaces at the steady-state. Retention of lactate is also linked to intra- and extracellular acidity, and hence a radial gradient of [HCO_3_^−^]. This gradient is notably smaller than the lactate gradient because CO_2_/HCO_3_^−^ is not the only acid-neutralizing buffer present inside cells; a sizable pool of buffering is provided by intrinsic buffers (15 mM/pH). The substantial build-up of intra- and extracellular lactate, and the depletion of HCO_3_^−^ at core regions is consistent with the retention of lactic acid in tumors in vivo [63]. This will likely inhibit glycolysis by the allosteric inhibitory effect of H^+^ ions of glycolytic enzymes (e.g., phosphofructokinase [64]), and trans-inhibition by lactate. Such a slowing of energy provision would be an Achilles heel to proliferating cancer cells. In response to this challenge, cancer cells are required to regulate their pH, and the canonical mechanism involves active transport [65]. An alternative means of reducing lactic acid build-up, without compromising metabolic rate, is to increase the diffusive capacity for venting lactate and HCO_3_^−^ ions.

## 4. Cancer Cells Can Exchange Lactate and Bicarbonate Ions through Gap Junctions

In the modeled scenario shown in Figure 2A, lactate (and HCO_3_^−^) can only diffuse radially along the extracellular path. In this system, the overall diffusivity for lactate is low, and a large radial gradient is needed to drive an efflux that will balance metabolic production. Transport capacity in the extracellular space is limited because this is the smaller of the two tissue compartments, and after accounting for tortuosity, the diffusion coefficients of small solutes, like lactate and bicarbonate, are of the order of ~10^3^ µm^2^/s. Lactate and bicarbonate ions are also transported in a chemically converted form (lactic acid and CO_2_) but this will be a smaller component of flux by >3 and >1 orders of magnitude, respectively. Any means of increasing effective diffusivity would benefit venting, and one way of achieving this is to engage in diffusion across the intracellular syncytium [35,36].

The substantial radial gradient of lactate in the intracellular space provides the energy to drive a dissipating flux, but only if there is sufficient coupling between cells (Figure 3A). On the basis of FRAP experiments, junctionally-coupled CRC and PDAC monolayers can have a permeability constant to calcein of the order of 0.03 µm^2^/s [36]. Anions such as lactate and bicarbonate are smaller than calcein (7- and 10-fold respectively), thus their permeability can be expected to be two orders of magnitude higher [38,39,66], i.e., in the order of ~2 and ~3 µm/s, respectively. The activity of MCT may produce lactic acid permeability as high as 10^3^ µm^2^/s, but this refers to the transport of lactic *acid*, which is >1000-times less abundant than its ionized conjugate. Lactate transport across the syncytial cytoplasmic volume could be described with an effective diffusion coefficient, D^eff^, given by Equation (3), where D is the diffusion coefficient between permeation barriers of permeability constant P, spaced at intervals d, i.e., approximated by cell diameter:(3)$$\frac{1}{D^{eff}}=\frac{1}{D}+\frac{1}{d\times P}$$

Small solutes like lactate may be expected to have effective diffusion coefficients in the cytoplasmic syncytium of ~30 µm^2^/s. Although this is more than an order of magnitude lower than diffusivity in the extracellular space, the ensuing intracellular flux can be meaningful because of the large volume of the intracellular compartment. Early studies have shown that low pH_i_ uncouples gap junctions [67,68], but more recent work has shown that gap junctions made of Cx43, for example, have a pH optimum that is slightly acidic relative to resting pH_i_ [69,70]; consequently, coupling by Cx43 gap junctions is expected to strengthen as pH_i_ falls modestly, before cells become uncoupled at unphysiologically low pH_i_.

Adding junctional lactate permeability to the mathematical model (Figure 3A) significantly improves lactic acid venting and reduces lactate build-up at the core (Figure 3B). However, the diffusive dissipation of lactate, a basic substance, will have an effect on pH [36]. At steady-state, lactate-only junctional efflux leaves behind H^+^ ions that acidify the intracellular space at the core of the tissue (Figure 3D). This generates a substantial radial gradient of [HCO_3_^−^] (Figure 3C).

To prevent this unwarranted acidification, junctions must also conduct HCO_3_^−^ ions, which would be expected to have a D^eff^, in this example, of 40 µm^2^/s. As shown in the results of a model with junctional lactate and HCO_3_^−^ permeability (Figure 4A), lactate build-up at the core can be reduced by a fifth (Figure 4B), without significantly acidifying the core (Figure 4C), thanks to a concurrent dissipation of any [HCO_3_^−^] gradient that develops in response to lactate transport.

## 5. Concluding Remarks

In this article, we describe a metabolic function for gap junctional coupling between cancer cells in solid tumors. Gradients of hypoxia in solid tumors also produce gradients of other solutes, such as lactate and HCO_3_^−^. These gradients are the driving forces for diffusion, but transport is only possible if there is adequate conductance. If this were to take place in the intracellular compartment of tumors, it must involve an adequate degree of coupling by gap junctions. In at least some cancers, such coupling is possible if connexin proteins are correctly targeted to points of cell-cell contact. Once a low-resistance pathway is established, it can lead to the diffusive exchange of solutes. Quantitatively the largest fluxes will involve those substances that have large gradients, such as lactate and HCO_3_^−^ anions. A lactate/HCO_3_^−^ exchange could benefit both of the coupled cells, as illustrated in Figure 5.

For the more hypoxic cell, gap junctions offer a route for lactate efflux, which reduces the retention of this glycolytic end-product. This was shown experimentally in spheroids of Colo357 cells, a Cx43-coupled PDAC cell line [36]. The recipient (more normoxic) cell can benefit from this lactate delivery by obtaining ATP from the mitochondrial respiration of this substrate [71]. Any excess lactate that is not respired can leave the normoxic cell by MCT, which is a means of removing H^+^ ions, another benefit to the recipient shown experimentally in PDAC spheroids [36]. Efflux of lactate, a base, from the hypoxic cell could potentially exacerbate its intracellular acidosis, but this is curtailed by HCO_3_^−^ delivery from the more alkaline normoxic cell, where HCO_3_^−^ is more plentiful. To sustain HCO_3_^−^ delivery, the normoxic cell can engage in active uptake, e.g., by Na^+^-driven HCO_3_^−^ cotransporters of the *SLC4* gene family [35]. From the viewpoint of the hypoxic cell, this active uptake mechanism is an example of remote secondary active transport that we described in PDAC spheroids [35]. Under this arrangement, the energy cost for delivering HCO_3_^−^ to the hypoxic cell is paid by the normoxic cell. In this example of barter, junctionally coupled cancer cells can benefit from the instantaneous and complete ionic trading.

## Figures and Tables

**Figure 1 cancers-11-00117-f001:**
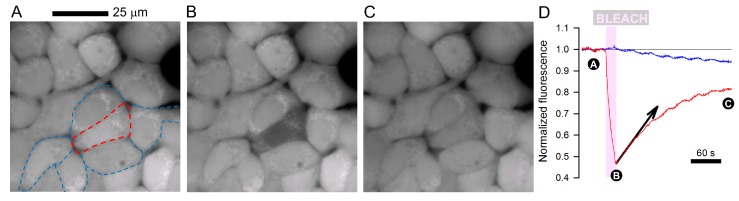
(**A**) Monolayer of SNU1235 colorectal cancer cells loaded with calcein (AM-ester; 7 min/4 µM) and imaged for fluorescence on a Zeiss LSM 700 confocal system (excitation 488 nm/emission > 510 nm). Red outline indicates cell that was selected for bleaching; blue outline highlights its direct neighbors. (**B**) Bleaching a central cell produces a gradient for calcein diffusion which can be tracked fluorescently. (**C**) Fluorescence in bleached cell recovers within 3 min, indicative of diffusion through gap junctions. (**D**) Recovery of fluorescence in the bleached cell is used to calculate junctional calcein permeability (Equation (2)), determined here to be 0.0491 µm/s. Note that fluorescence in neighboring cell decreases as signal from the bleached cell recovers.

**Figure 2 cancers-11-00117-f002:**
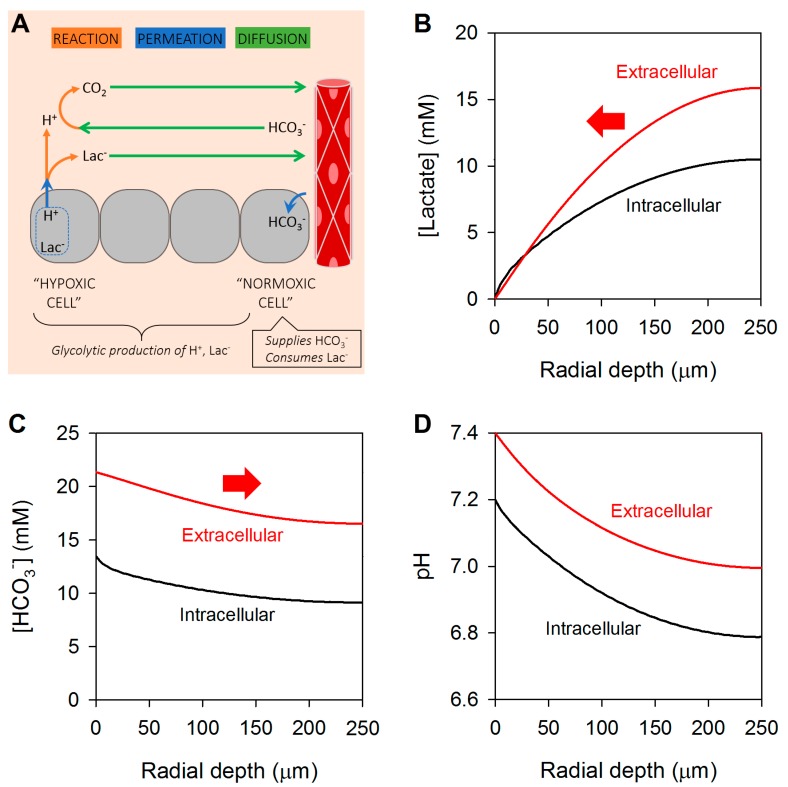
Modeling the radial distribution of lactate, HCO_3_^−^ and H^+^ ions in tissues without junctional coupling, i.e., radial diffusion in extracellular spaces only. (**A**) Schematic representation of the 3-D spherical geometry representing a solid tumor with a hypoxic core. Results of modelling shown as radial concentration gradients at the steady-state for (**B**) lactate, (**C**) HCO_3_^−^ and (**D**) pH in the extracellular (red) and intracellular spaces (black). Arrows indicate direction of diffusive flux.

**Figure 3 cancers-11-00117-f003:**
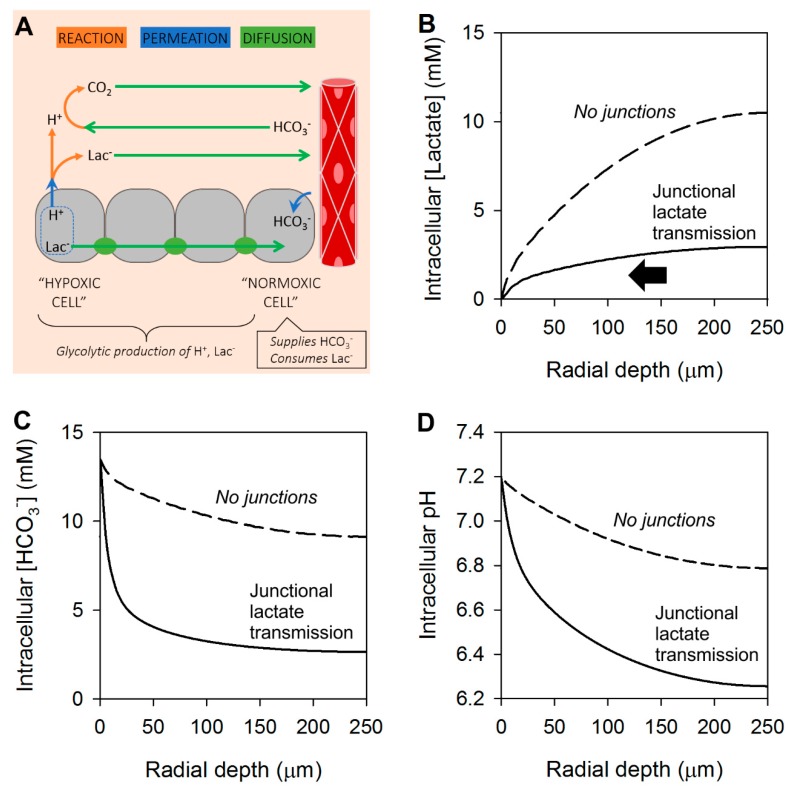
Modeling the radial distribution of lactate, HCO_3_^−^ and H^+^ ions in tissues with junctional coupling for lactate only. (**A**) Schematic representation of the 3-D spherical geometry representing a solid tumor with a hypoxic core. Results of modelling shown as radial concentration gradients at the steady-state for (**B**) lactate, (**C**) HCO_3_^−^ and (**D**) pH in the intracellular space. Arrows indicate direction of diffusive flux. For comparison, dashed lines show results with no junctional coupling, from Figure 2. Note that diffusion of lactate and lactic acid can take place across the intra- and extracellular spaces. Junctional efflux of lactate (the major permeating species) leaves behind H^+^ ions, and hence a depletion of HCO_3_^−^; this generates a large radial gradient of [HCO_3_^−^].

**Figure 4 cancers-11-00117-f004:**
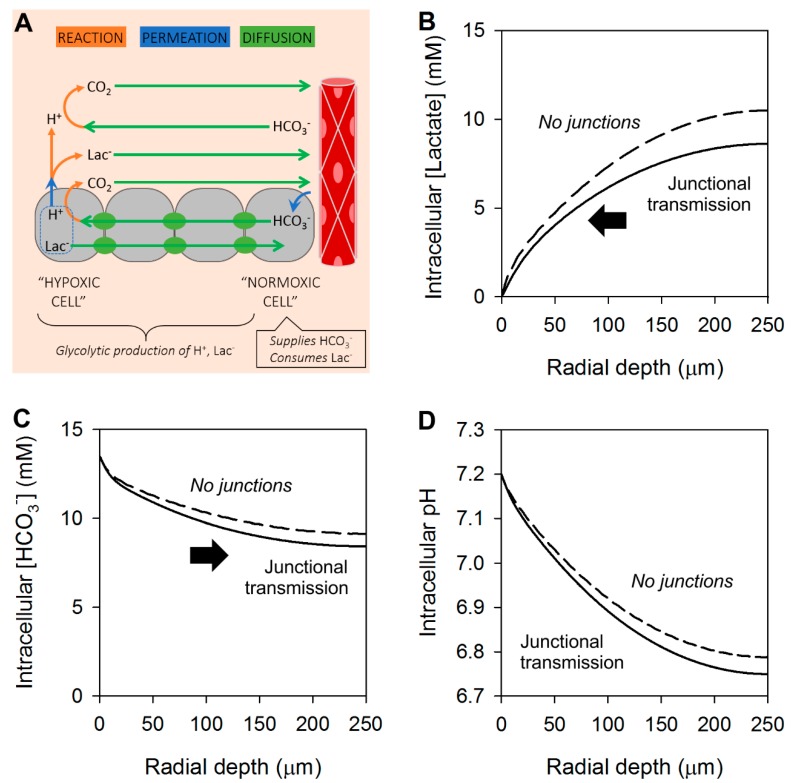
Modeling the radial distribution of lactate, HCO_3_^−^ and H^+^ ions in tissues with junctional coupling for all ionic species. (**A**) Schematic representation of the 3-D spherical geometry representing a solid tumor with a hypoxic core. Results of modelling shown as radial concentration gradients at the steady-state for (**B**) lactate, (**C**) HCO_3_^−^ and (**D**) pH in the intracellular space. Arrows indicate direction of diffusive flux. For comparison, dashed lines show results with no junctional coupling, from Figure 2. In this system, intracellular lactate diffuses radially out of the core, leaving behind H^+^ ions. This generates an inverse gradient of intracellular [HCO_3_^−^], which drives a counter-flux of HCO_3_^−^ i.e., junctional lactate/HCO_3_^−^ exchange. Overall, lactate venting improves by a fifth.

**Figure 5 cancers-11-00117-f005:**
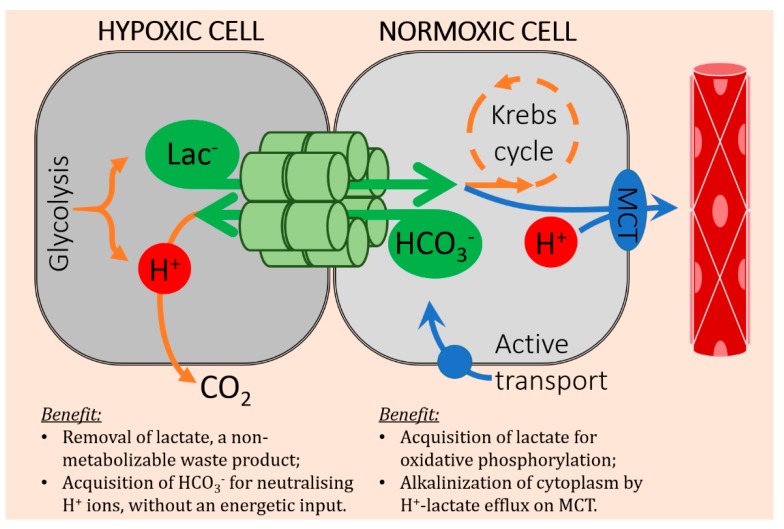
Schematic diagram of junctional lactate/HCO_3_^−^ exchange between two coupled cancer cells, one of which is more hypoxic i.e., further away from a capillary. The barter of lactate for HCO_3_^−^ can benefit both partners.

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
