# Peer review of "A Barter Economy in Tumors: Exchanging Metabolites through Gap Junctions"

_cancers, 2019, doi:10.3390/cancers11010117_

Reviewer 1 Report

The authors present a very intriguing hypothesis on the role of gap junctions in the exchange of lactate from cancer cells within the more hypoxic center of tumors with that of bicarbonate from the more normoxic cells adjacent to capillaries.  In the latter stages of cancer progression, tumors become more heterogeneous in cell types with connexins being expressed at some stages in subpopulations of cells in some tumors.  The expression and re-establishment of some intercellular communication will allow a division of labor among cells in the tumor to exchange the lactate being produced in the hypoxic cells from glycolysis to the more normoxic cells near capillaries where it can be metabolized in mitochondria and co-transported with H+ outside of the cells via the H+-monocarboxylate transporter.  Mathematical models on the radial distribution of bicarbonate, lactate and H+ as a function of gap junction permeability are presented in three figures.  The hypothesis is clearly presented and made very interesting, particularly with the barter economy analogies.  This work definitely falls into the scope of the journal’s special issue on gap junctions, advances our scientific knowledge in this field.  I recommend publication after consideration of the following comments and concerns

The authors should indicate in the beginning of the abstract that this is a hypothetical modeling study.

More details are needed in modeling the data used to generate the graphs of Figurers 2, 3 and 4. Was the diffusivity equation 3 used?  What program was used to generate the data?  What were the assumptions?  Were the assumptions for junctional transmission data based on channels completely opened.  Also discuss some of the limitations of the model.

Does Figure 1 have any direct relations to the modelled data, or was this figure to demonstrate one method of measuring intercellular communication, thus maybe not all that important to this paper.  Also, what instrument was used to make these measurements.

Author Response

We thank the Reviewer for the helpful comments on how to improve our manuscript.  We have addressed the Reviewer’s comments as follows.

Point 1: The authors should indicate in the beginning of the abstract that this is a hypothetical modeling study.

The abstract now clarifies that this is a computational modelling study, based on measurements obtained by us in two previous studies.

Point 2: More details are needed in modeling the data used to generate the graphs of Figurers 2, 3 and 4. Was the diffusivity equation 3 used?  What program was used to generate the data?  What were the assumptions?  Were the assumptions for junctional transmission data based on channels completely opened.  Also discuss some of the limitations of the model.

We agree that the details of the modelling are missing. To maintain the flow of the text, we have added an Appendix that describes the model equations, lists our choices of variables and justifies the assumptions we have made, and its limitations.

Point 3: Does Figure 1 have any direct relations to the modelled data, or was this figure to demonstrate one method of measuring intercellular communication, thus maybe not all that important to this paper.  Also, what instrument was used to make these measurements?

We apologize for not clarifying the purpose of this figure.  We included these exemplar measurements to illustrate how junctional permeability could be measured in cancer cell monolayers.  In Fig1 legend, we provide further methodological details, including equipment.

Reviewer 2 Report

Interesting commentary article that discusses one of the possible beneficial roles of gap junctional coupling in some solid tumors which is to enhance the exchange of lactate and bicarbonate ions between the hypoxic cancer cells and their normoxic neighbors for the purpose of supporting an elevated metabolic rate. Well written.  Results of mathematical model support the main hypothesis of the paper.

Author Response

Interesting commentary article that discusses one of the possible beneficial roles of gap junctional coupling in some solid tumors which is to enhance the exchange of lactate and bicarbonate ions between the hypoxic cancer cells and their normoxic neighbors for the purpose of supporting an elevated metabolic rate. Well written.  Results of mathematical model support the main hypothesis of the paper.

We thank the Reviewer for the encouraging comments. Please note we have revised the manuscript to address questions from another Reviewer (to expand on the methods and modeling approach).